# Glycyrrhiza uralensis promote the metabolism of toxic components of Aconitum carmichaeli by CYP3A and alleviate the development of chronic heart failure

Lulu Ni[1☯], Ping Miao[2☯], Jian Jiang[3☯], Fang Wan[4], Jiangan Li[4], Min Ai[1], Lingzhong Kong[5], Su Tu[4]*

1 Department of Basic Medicine, Jiangnan University, Wuxi, China, 2 Traditional Chinese Medicine Diagnosis and Treatment Center, The Affiliated People's Hospital of Ningbo University, Zhejiang, China, 3 Department of Clinical Pharmacology, Shuguang Hospital Affiliated to Shanghai University of Traditional Chinese Medicine, Shanghai, China, 4 Department of Emergency, The Affiliated Wuxi NO.2 People's Hospital of Nanjing Medical University, Wuxi, PR China, 5 Department of Rehabilitation Acupuncture Medicine, Bozhou People's Hospital, Bozhou, Anhui, PR China

☯ These authors contributed equally to this work.
* tusuwxey@126.com

**Data Availability Statement:** All relevant data are within the manuscript and its Supporting Information files.

## Abstract

Aconitum, as "the first drug of choice for invigorating Yang and saving lives", has been widely used for the treatment of heart failure. However, toxic components of Aconitum can easily lead to serious arrhythmia, even death (Y. CT., 2009; Zhang XM., 2018). In this study, a High Performance Liquid Chromatography (HPLC) method for the determination of aconitine (AC), mesaconitine (MA) and hypaconitine (HA) was established; The effect of Glycyrrhiza on CYP3A1 / 2 mRNA expression was detected by RT-PCR; SD rats were given Aconitum and compatibility of Glycyrrhizae and Aconitum by gavage respectively, the blood concentration of toxic components were determined by LC-MS / MS; The CHF rat model was established by intraperitoneal injection of adriamycin (2.5 mg / kg), and were randomly divided into model, Aconitum, the compatibility of Glycyrrhizae and Aconitum and Captopril group, 5 mice/group. After 4 weeks of gavage, the corresponding indexes were detected by ELISA and HPLC. The results showed that Ketoconazole significantly inhibited the metabolites of AC, MA and HA; Glycyrrhiza induced CYP3A gene expression; The level of ALD in the compatibility of Glycyrrhizae and Aconitum group was significantly lower than that in Aconitum group. After intervention with the compatibility of Glycyrrhizae and Aconitum, ATP increased, ADP decreased significantly. In conclusion, we found Glycyrrhiza promoted the metabolism of toxic components of Aconitum by up regulating the expression of CYP3A, and reduced the content of BNP, Ang II and ALD, improved the energy metabolism disorder of myocardium, alleviated the development of CHF.

**Funding:** This study was supported by National Natural Science Foundation of China (81904171), Jiangsu Postdoctoral Research Foundation (2020Z388), Top Talent Support Program for young and middle-aged people of Wuxi Health Committee, General fund of Wuxi health committee (M202033).

**Competing interests:** The authors have declared that no competing interests exist.

## Introduction

CHF is a common clinical critical disease, which is the end stage of heart disease caused by various causes, with high mortality and short survival [1]. The treatment of CHF with TCM is increasingly recognized by the medical community and patients, TCM plays an important role in the prevention and treatment of heart failure [2]. Aconitum is the root products of Aconitum carmichaeli Debx. as "the first drug of choice for invigorating Yang and saving lives", it has been widely used for the treatment of heart failure. Many components of Aconitum (such as noraconitine, aconitine glycosides, alkanolamine alkaloids, etc.) have positive inotropic [3, 4], cardiotonic, pressor, anti shock, analgesic and anti-inflammatory effects [5]. However, diester diterpenoid alkaloids (AC, MA and HA) are toxic components of Aconitum, which can easily lead to serious arrhythmia, respiratory depression, shock and even death [6, 7]. In traditional medicine, Glycyrrhizae combined with Aconitum is a representative drug pair for reducing toxicity and increasing efficiency. In recent years, researchers have done a lot of research on Glycyrrhiza [5, 8–12]. Glycyrrhiza is the dry roots and rhizomes of Glycyrrhiza uralensis Fisch, Glycyrrhiza inflata Bat, or Glycyrrhiza glabra L., the main effective components are triterpenoid saponins and flavonoids. Glycyrrhiza can obviously antagonize arrhythmia [13]. Zhang et al. [12] found that the antiarrhythmic mechanism of glycyrrhetinic acid may be related to the opening of L-type calcium channel and the increase of intracellular calcium concentration.

Cytochrome CYP450, which is mainly distributed in the liver, is the most important drug metabolizing enzyme in human body and plays an important role in the metabolism of a variety of endogenous and exogenous substances [14]. The inhibition or induction of drug metabolizing enzymes by many TCM and their components is one of the most common causes of herb-herb interaction [15], which affects the metabolic process of other drugs combined with them. Thus it can be seen that CYP450 is of great significance in the study of compatibility of TCM. In this study, the metabolism of AC, MA and HA in Aconitum was intervened by regulating CYP450 with Glycyrrhiza, and the protective effect and mechanism of Glycyrrhiza combined with Aconitum on CHF were further explored from the perspective of neuroendocrine system and myocardial energy metabolism in rats with heart failure, so as to avoid the adverse reactions caused by Aconitum alone.

## Materials and methods

### Experiment reagents

AC was purchased from Shanghai Yuanye Biotechnology Co., Ltd; MA, HA, quinidine and ketoconazole were purchased from China Institute for the control of pharmaceutical and biological products; Furatheophylline, sulfabendazole and reduced NADPH were purchased from Sigma, USA; Rat liver microsomes were purchased from Research Institute for Liver Disease, Shanghai; Methanol and acetonitrile are from Tedia, USA; ADR was purchased from Haizheng Pharmaceutical, Zhejiang, China; Captopril tablets were purchased from Squibb Pharmaceutical, Shanghai; BNP, ANG II and ALD detection kit were all provided by Westang biotech, Shanghai; AMP Na2, ADP Na2 and ATP Na2 were purchased from Amresco, USA; Fluorescent quantitative PCR kit was purchased from Takara, China. Glycyrrhizae and Aconitine were purchased from LEIYUNSHANG, Shanghai, they were boiled and concentrated and stored in refrigerator at 4˚C.

Agilent 1200 high performance liquid chromatograph (Agilent company, USA), equipped with four element gradient pump (G1311A), online degasser (A1311A), automatic sampler (G1329A), column incubator (G1314B), UV detector (G1314B) and Agilent chemical

workstation; ABI 4000 QTrap mass spectrometer (ABI / SCIEX, USA); Eckman Allegra 64R high-speed freezing centrifuge (Beckman, USA); ABI viiA7 real-time fluorescent quantitative PCR (ABI, USA); Optima TM LE-80K ultracentrifuge (Beckman, USA); Avanti J-E multi-purpose high efficiency centrifuge (Beckman company, USA).

## Chromatographic conditions

Agilent C18 column (4.60 mm × 150 mm, 5μm), mobile phase A was 20 mmol / L ammonium acetate and 0.1% formic acid aqueous solution, mobile phase B was methanol, isocratic elution, A: B = 60:40, flow rate was 1 ml / min, detection wavelength was 240 nm, column temperature was 30˚C, injection volume was 20 μL.

## Determination of the metabolism of AC, MA and HA by HPLC

1 mg / ML microsome 100 μL, PBS 75 μL and aconitine alkaloids 5 μL (AC, MA and HA: 1 mmol / L) were added. 20 μL NADPH (10 mmol / L) was added to start the reactionand, AC and HA were incubated for 90 min, MA were incubated for 120 min. The reaction was terminated by adding 200 μL glacial acetonitrile, 12 000 r / min at 4˚C for 10 min, and 20 μL supernatant was used for HPLC.

## The effect of CYP450 inhibitor on metabolism of AC, MA and HA

Added microsome 100 μL, PBS 75 μL, aconitine alkaloids 5 μL, inhibitors (furatheophylline: 0.2 mmol / L, sulfabendazole: 0.4 mmol / L, quinidine: 0.1 mmol / L, ketoconazole: 0.04 mmol / L) 10 μL. The operation of HPLC was as above. The effect of CYP450 inhibitors on the metabolism of Aconitum alkaloids was calculated. The formula is as follows:

% inhibition = Peak area of X after adding inhibitor / Peak area of X before adding inhibitor × 100%, where X is metabolites of AC, MA and HA.

## Effects of different doses of Glycyrrhiza on CYP3A1/2 expression in rats

All animal procedures, including killing, were approved by the Institutional Animal Care and Use Committee at Medical College of Shanghai University of TCM. 25 SD rats, weighing 180–200 g, were purchased from Shanghai SLAC Laboratory Animal company and fed in SPF environmental condition. The rats were randomly divided into negative control, phenobarbital positive induction (0.08 g / kg / D) and Glycyrrhiza groups with different concentrations (0.33 g / kg / D, 1 g / kg / D and 3 g / kg / D), with 5 rats in each group, intragastric administration for 14 days. The phenobarbital induced group was given intraperitoneal injection three days before treatment. The animals were killed by CO2 asphyxiation, liver microsomes were prepared and were used to detect the expression of CYP3A1 / 2 mRNA in liver tissue.

## Preparation of rat liver microsomes

Each gram of liver tissue was added with 4 mL buffer solution, homogenized in ice bath, 4˚C and 12000 g for 15 min. The supernatant was centrifuged at 105000 g at 4˚C. After 70 minutes, the buffer solution was suspended and precipitated, and centrifuged with 105000g at 4˚C for 45min. 1 mL of 0.25 mol / L sucrose was added to every 4 g of liver tissue, which was gently blown and resuspended, stored at—80˚C.

## Real-time PCR

Total RNA was extracted using Trizol reagent according to the manufacturer's instructions. 1μg of total RNA was transcribed to cDNA using the PrimeScriptTM RT Master Mix kit

(TaKaRa, China). Quantitative RT-PCR was performed in a reaction volume of 20μL cDNA on ABI system (Applied Biosystems, Life Technologies). The average cycle thresholds (Ct) were employed to quantify fold-change. The $2^{-\triangle\triangle CT}$ method was reported to calculate relative gene expression levels. The Primers for all genes were listed as follows: GAPDH forward: TGAGGTGACCGCATCTTCTTG; GAPDH reverse: TGGTAACCAGGCGTCCGATA; CYP3A1 forward: GATGTTGAAATCAATGGTGTGT; CYP3A1 reverse: TTCAGAGGTATCTGTGTTTCC; CYP3A2 forward: AGTAGTGACGATTCCAACATAT; CYP3A2 reverse: TCAAGAGTATCTGTGTTTCCT.

## The determination of AC, MA and HA in blood after Glycyrrhizae with Aconitum

12 SD rats were randomly divided into Aconitum and Glycyrrhizae combind with Aconitum group, 6 rats in each group, 2 mL / kg was given by gavage respectively. At 10, 20, 30, 45, 60, 90, 120, 240, 360, 480, 720 and 1440 min after administration, 0.3 mL of blood was taken from the ophthalmic venous plexus for LC-MS / MS to detect the contents of AC, MA and HA in blood.

## Establishment of rat CHF model with ADR

According to reference [16], the rat model of CHF was established by intraperitoneal injection of ADR (2.5 mg / kg) once a week for six weeks. 6 weeks later, the rats were randomly divided into model group, Aconitum (0.3 g / mL), Glycyrrhizae combind with Aconitum (0.6 g / mL) and Captopril (1.25 mg / mL) group. Each treatment group was given 5 mL / kg / D by gavage for 4 weeks, the normal and model group were given the same volume of distilled water by gavage. After the last administration, the rats were killed by CO2 asphyxia, and the indexes were determined.

## ELISA determined the levels of BNP, Ang II and ALD in serum

ELISA assays were performed to measure BNP, Ang II and ALD levels according to the manufacturer's protocols. The cells in each sample were collected and stored—20˚C prior to use. The standards or samples diluent were distributed into each well and the plate incubated for 1 h at 37˚C. The reacted plate was rinsed and then incubated for 30 min at 37˚C with indicated antibodies and then re-washed. The reactions were terminated with 50 mL stop solution, and the optical density (OD) at 450 nm was measured with a microplate reader (STNERGY/H4, BioTek, Vermont, USA).

## Determination of ATP, ADP and AMP in tissues by HPLC

Weighed 100 mg of left ventricular myocardium of rats, cut into pieces, added 500 μL 4.2% perchloric acid, homogenized in ice bath, 3 000 r / min, 4˚C for 15 min. Took 200 μL of the supernatant, added 135 μL of 1 mol / L NaOH, adjusted the pH to about 6.0, shaked and mixed, 8 000 r / min, 4˚C for 10 min, and took 20 μL of the supernatant for HPLC analysis. There was a good linear relationship between the peak area measured by HPLC and the concentrations of ATP, AMP and ADP, and the content was calculated according to the standard curve.

## Statistical analysis

All data were statistically analysed using GraphPad Prism software version 5.0 (GraphPad Software, Inc. La Jolla, USA). The data are represented as the mean ± standard error of mean

(SEM) of three independent biological experiments (n = 3 in vitro, n = 5 in vivo). Comparisons between treatment groups were performed using one-way analysis of variance (ANOVA), followed by a student's t-test. A p-value $\leq$ 0.05 was considered as statistically significant.

## Result

### Determination of AC, MA, HA and their metabolites by HPLC

Under the experimental conditions, there was no impurity peak interference in the blank rat liver microsomes (Fig 1A). AC produced three metabolites (A1-A3) in RLMs with retention times of 29.10 min, 15.05 min, 18.53 min and 21.88 min, respectively (Fig 1B). The retention times of MA and its metabolites M1, M2 and M3 were 20.41 min, 11.01 min, 13.20 min and 13.83 min, respectively (Fig 1C). HA produced two metabolites with retention times of 26.44 min, 13.26 min and 18.35 min, respectively (Fig 1D).

### Effects of incubation time on metabolites of AC, MA and HA

The metabolites of AC and HA increased with the increase of incubation time after 30 min, 60 min and 90 min. At 120 min, the production of metabolites basically reached the plateau stage, the optimal incubation time was 90 min (Fig 2A and 2C). The amount of metabolites of MA increased linearly in the incubation of 30–120 min (Fig 2B).

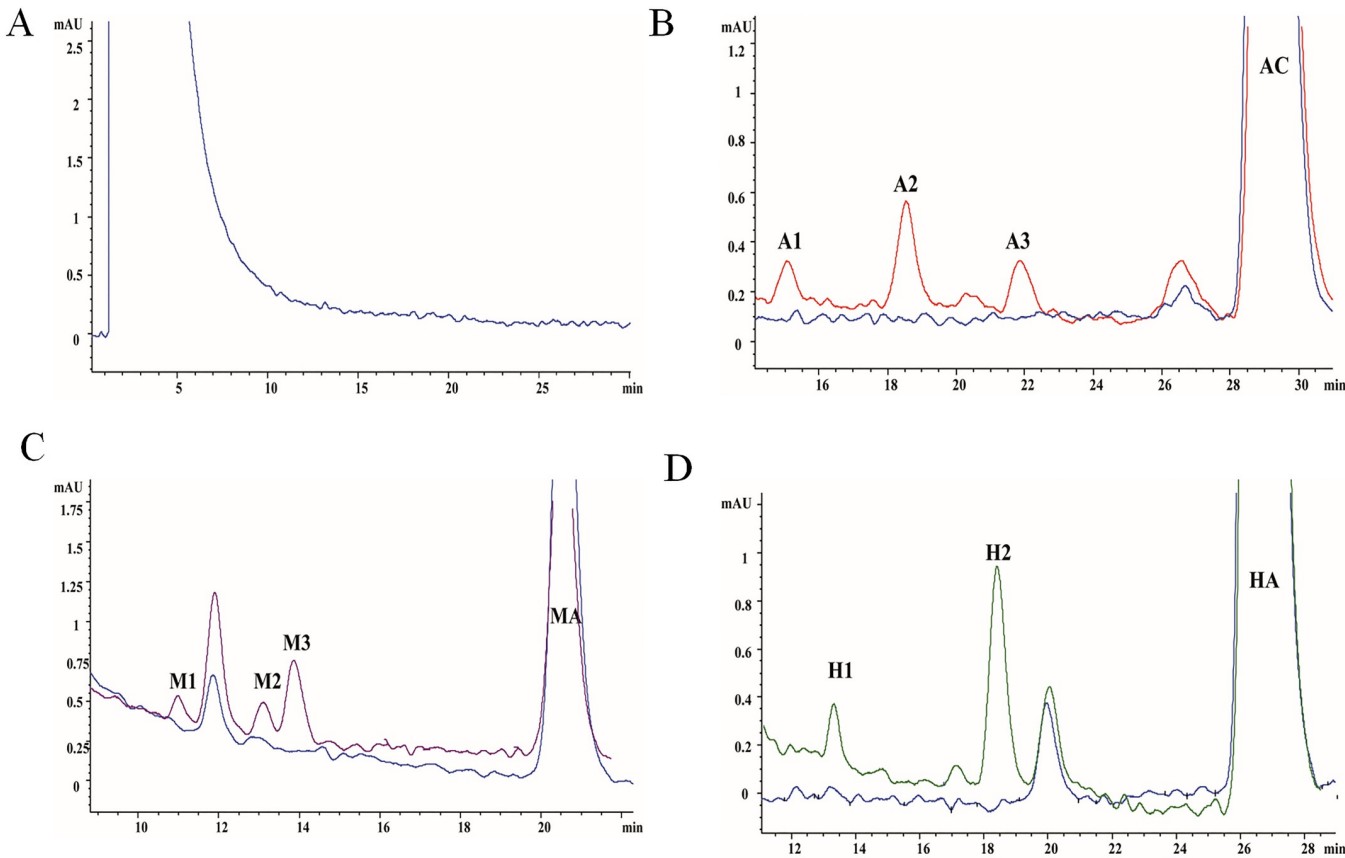

**Fig 1. HPLC chromatogram of toxic components and metabolites of Aconitum in Rat Liver Microsomes (RLM).** 1 mg / ml microsome 100 μL, PBS 75 μL and aconitine alkaloids 5 μL (1 mmol / L) were added in turn, AC and HA were incubated for 90 min, MA were incubated for 120 min for HPLC respectively. (A) HPLC chromatograms of blank microsome sample. (B) HPLC chromatograms of AC and its metabolites in RLMs. (C) HPLC chromatograms of MA and its metabolites in RLMs. (D) HPLC chromatograms of HA and its metabolites in RLMs.

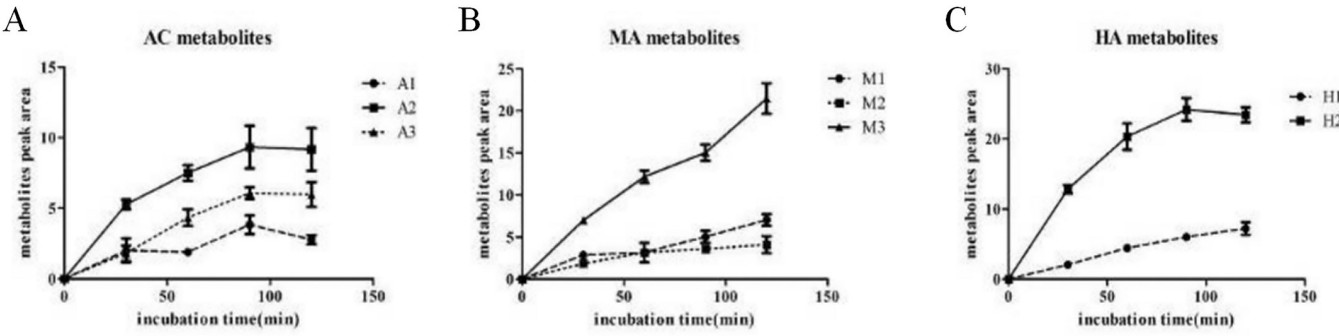

**Fig 2. Time-peak area graph of the metabolites of AC, MA and HA in RLMs.** 1 mg / ml microsome 100 μL, PBS 75 μL and aconitine alkaloids 5 μL (1 mmol / L) were added in turn, AC and HA were incubated for 90 min, MA were incubated for 120 min for HPLC respectively. (A) The metabolites of AC increased with the increase of incubation time after 30 min, 60 min and 90 min. At 120 min, the production of metabolites basically reached the plateau stage, the optimal incubation time was 90 min. (B) The amount of metabolites of MA increased linearly in the incubation of 30–120 min. (C) The amount of metabolites of HA increased linearly in the incubation of 30–90 min, at 120 min, the production of metabolites reached the plateau stage.

### Intervention of CYP450 inhibitors on metabolism of AC, MA and HA

The results showed that ketoconazole, a specific inhibitor of CYP3A, could significantly reduce the production of AC metabolites (A1, A2, A3) (-0.2416 ± 0.0644; -0.4708 ± 0.02905; -0.4643 ± 0.06743; $P < 0.05$; $P < 0.001$; $P < 0.01$), MA metabolites (M1, M2, M3) (-0.4085 ± 0.07279; -0.3312 ± 0.06126; -0.3905 ± 0.01824; $P < 0.01$; $P < 0.01$; $P < 0.001$) and HA metabolites (H1, H2) (-0.4482 ± 0.05543; -0.4161 ± 0.01634; $P < 0.01$; $P < 0.001$). Other inhibitors could also inhibit the production of HA metabolites (H2) (0.09065 ± 0.02734; -0.1343 ± 0.01908; -0.1179 ± 0.0223; $P < 0.05$; $P < 0.01$; $P < 0.01$), but furaphylline had no significant effect on the metabolism of AC, MA and HA (Fig 3).

### Effects of Glycyrrhiza on CYP3A mRNA expression and metabolism of AC, MA and HA

Previous studies found that CYP3A was the main metabolic enzyme of AC, MA and HA. Then we look at the effect of Glycyrrhiza on CYP3A mRNA expression, and the effect of Glycyrrhiza on the metabolism of diester alkaloids in vitro, in order to explore the mechanism of Glycyrrhiza promoting the metabolism of toxic components of Aconitum from the perspective of drug metabolizing enzyme CYP3A. Compared with the negative control group, different concentrations of Glycyrrhiza all could significantly induce the expression of CYP3A 1 / 2 mRNA (CYP3A1: 1.889 ± 0.2166; 2.122 ± 0.2511; 2.485 ± 0.2511; $P < 0.01$; $P < 0.01$; $P < 0.001$; CYP3A2: 1.341 ± 0.107; 1.417 ± 0.09821; 1.79 ± 0.1269; $P < 0.001$; $P < 0.001$; $P < 0.001$), but there was no significant difference among low, medium and high concentrations of Glycyrrhiza (Fig 4). Compared with the negative control group, the metabolism rate of AC, MA and HA in the liver of Glycyrrhiza low and middle dose groups were significantly increased after co-incubation of Aconitum diester alkaloids and rat liver microsomes which induced by Glycyrrhiza (Fig 5).

### Glycyrrhizae combined with Aconitum reduced the blood concentration of AC, MA and HA in rats

On the basis of previous work, the dynamic changes of AC, MA and HA in Aconitum and in compatibility of Glycyrrhizae and Aconitum were detected by LC-MS / MS, which is helpful to clarify the scientificity and rationality in compatibility of Glycyrrhizae and Aconitum. It was

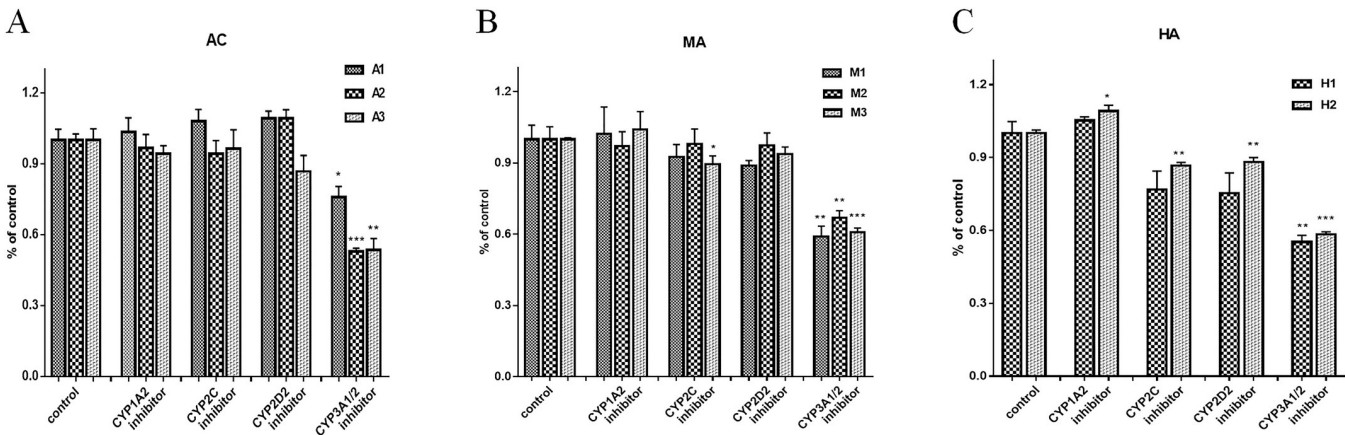

**Fig 3. Effects of CYP inhibitors on the formation of AC, MA and HA metabolites.** (A) AC; (B) MA; (C) HA. Compared with the control, $^*P<0.05$; $^{**}P<0.01$; $^{***}P<0.001$.

found that the plasma concentrations of AC (Fig 6A), MA (Fig 6B) and HA (Fig 6C) in rats with compatibility of Glycyrrhizae and Aconitum were significantly lower than those with Aconitum alone.

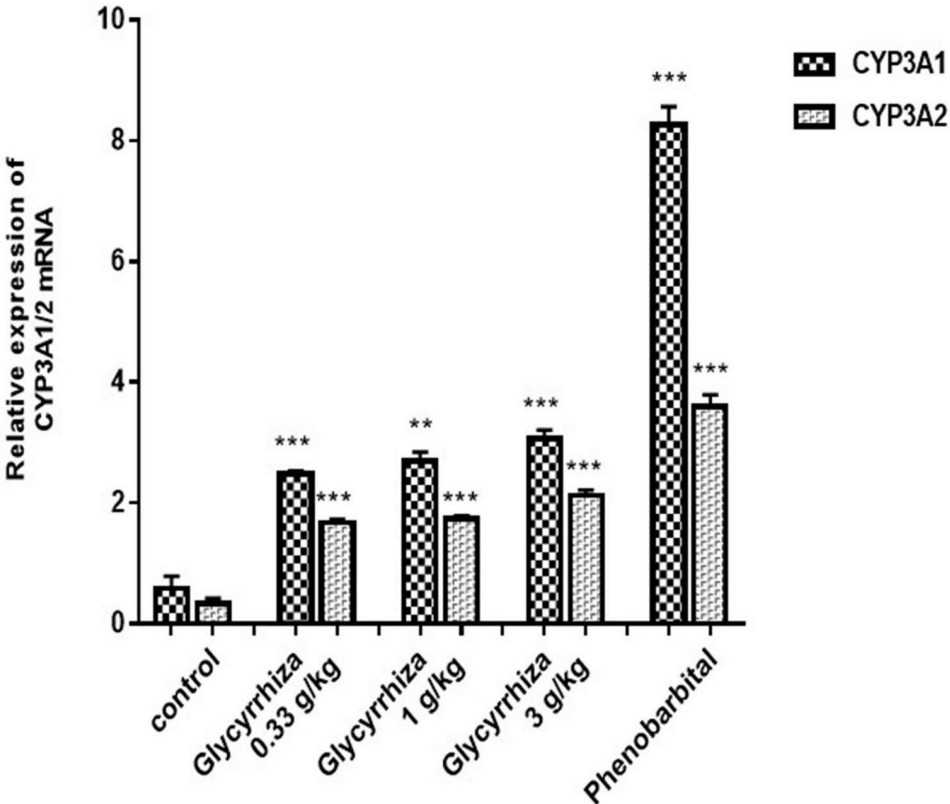

**Fig 4. Effect of Glycyrrhizae on CYP3A1/2 mRNA expression.** Compared with the negative control group, different concentrations of Glycyrrhiza all could significantly induce the expression of CYP3A 1 / 2 mRNA, $^{**}P<0.01$; $^{***}P<0.001$.

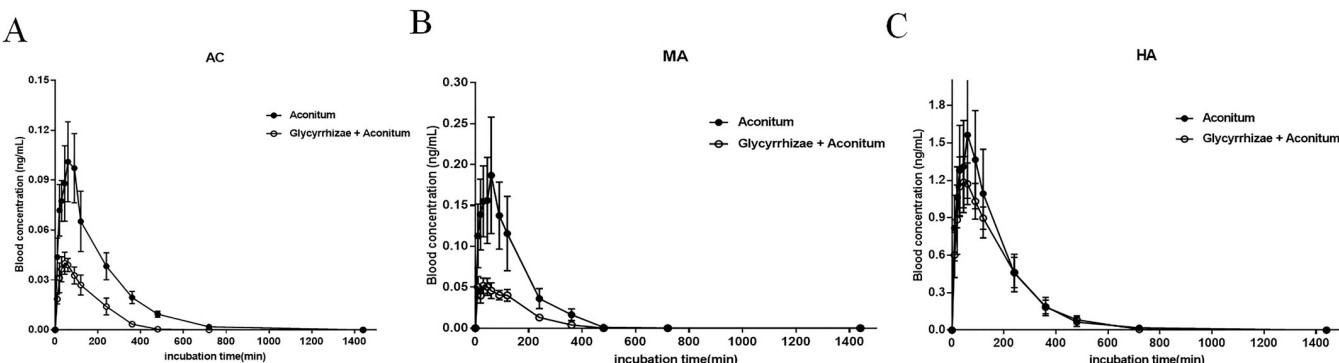

**Fig 5. Effects of different concentrations of Glycyrrhiza on the production of toxic metabolites A1, A2, M1 and H1 in Aconitum.** Compared with the negative control group, the metabolism rate of AC, MA and HA in Glycyrrhiza low and middle dose groups were significantly increased.

**Fig 6. The effects of Aconitum and the compatibility of Glycyrrhizae and Aconitum on the plasma concentrations of AC, MA and HA in rats.** Aconitum and the compatibility of Glycyrrhizae and Aconitum 2 mL / kg were given by gavage respectively. At 10, 20, 30, 45, 60, 90, 120, 240, 360, 480, 720 and 1440 min after administration, 0.3 mL of blood was taken from the ophthalmic venous plexus, separated plasma and analyzed by LC-MS / MS. The plasma concentrations of AC (A), MA (B) and HA (C) in rats with compatibility of Glycyrrhizae and Aconitum were significantly lower than those with Aconitum alone.

## Glycyrrhiza combined with Aconitum reduced BNP, AngII and ALD levels in serum

In this study, ADR was used to establish the rat model of CHF. Compared with the normal group, the BNP content in the model group was significantly increased (6045 ± 606.7; $P < 0.001$). Compared with the model group, the BNP level in each treatment group was decreased (-4178 ± 680.6; -4532 ± 666.1; -5783 ± 677; $P < 0.001$; $P < 0.001$; $P < 0.001$). But there was no significant difference between Aconitum group and the compatibility of Glycyrrhizae and Aconitum group (Fig 7A). The content of AngIIin serum of CHF model group was significantly higher than that of normal group (12.43 ± 3.226; $P < 0.01$), and the treatment groups were significantly lower than that of model group (-10.38 ± 2.871; -10.96 ± 3.18; $P < 0.01$; $P < 0.01$) (Fig 7B). Compared with the normal group, the content of ALD in serum of the model group was significantly increased (600 ± 65.51; $P < 0.001$), and the ALD in each drug group was significantly decreased after intervention (-360.1 ± 87.42; -505.4 ± 54.42; -581.3 ± 58.86; $P < 0.01$; $P < 0.001$; $P < 0.001$). Among them, the level of ALD in the compatibility of Glycyrrhizae and Aconitum group was lower than that in Aconitum group (-145.4 ± 56.8; $P < 0.05$) (Fig 7C).

## Changes of adenosine monophosphate content in rat myocardium after intervention of the compatibility of Glycyrrhizae and Aconitum

Compared with the normal group, the ATP content in CHF model group was decreased (-0.09453 ± 0.009412; $P < 0.001$); compared with the model group, the ATP content in

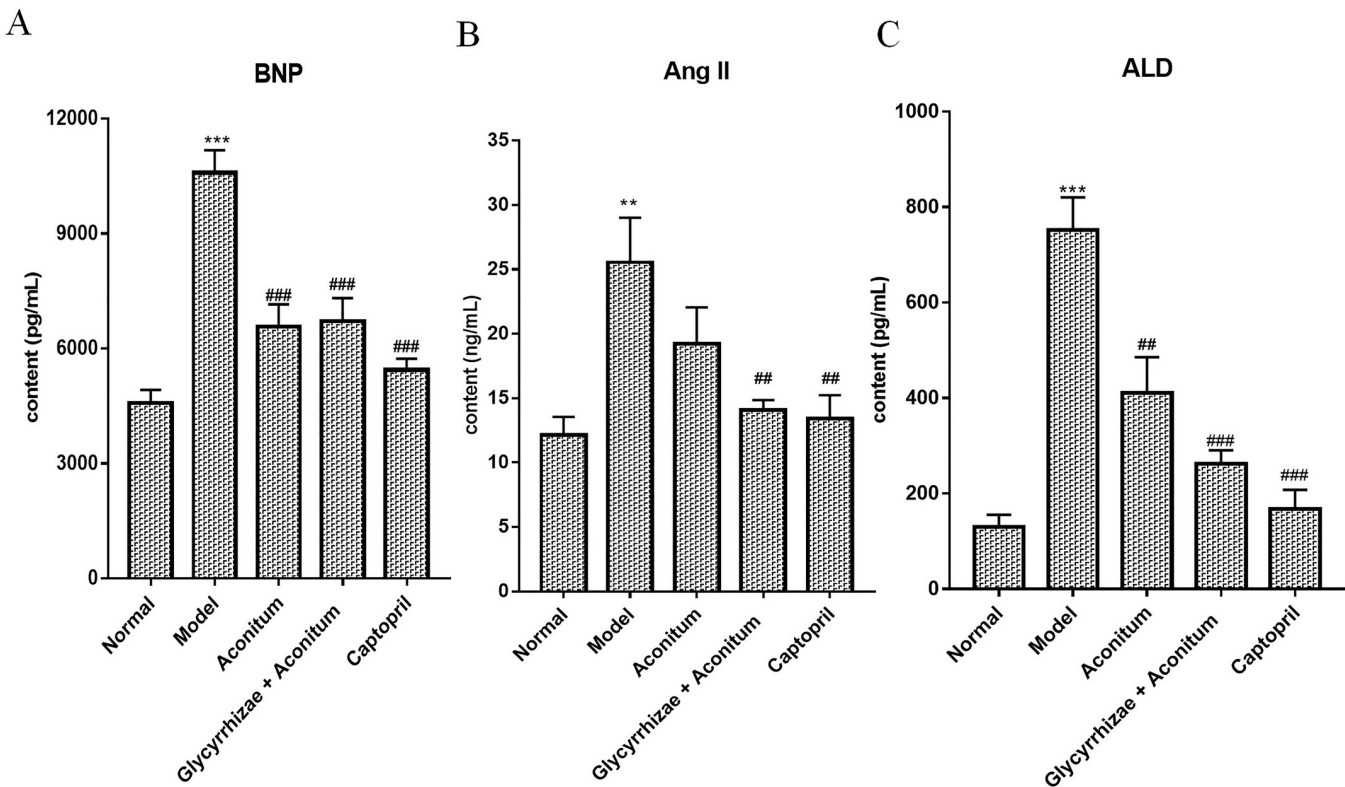

**Fig 7. The intervention effect of Aconitum and the compatibility of Glycyrrhizae and Aconitum on serum BNP, ang II and ALD levels.** The rat model of CHF was established by intraperitoneal injection of ADR (2.5 mg / kg). The rats were randomly divided into model group, Aconitum (0.3 g / mL) group, Glycyrrhizae with Aconitum (0.6 g / mL) group and Captopril (1.25 mg / mL) group. Each treatment group was given 5 mL / kg / D by gavage for 4 weeks, the normal group and model group were given the same volume of distilled water by gavage. (A) The levels of BNP in serum determined by ELISA. (B) The levels of Ang II in serum determined by ELISA. (C) The levels of ALD in serum determined by ELISA. Compared with the normal group, **P<0.01; ***P<0.001; Compared with the model group, ##P<0.01; ###P<0.001; Compared with Aconitum group, $ P<0.05; $ $ P<0.01.

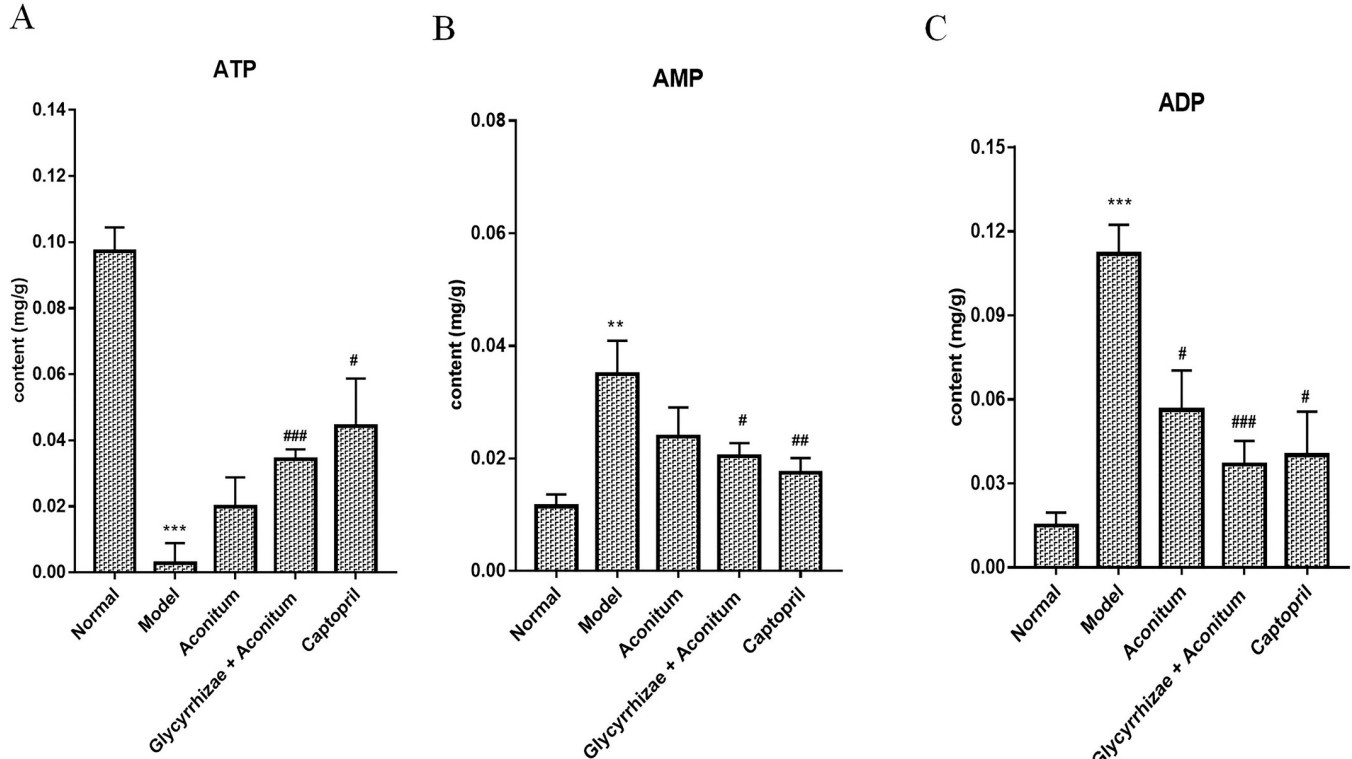

**Fig 8. The intervention effect of Aconitum and the compatibility of Glycyrrhizae and Aconitum on serum ATP, AMP and ADP levels.** The rat model of CHF was established by intraperitoneal injection of ADR (2.5 mg / kg). The rats were randomly divided into model group, Aconitum (0.3 g / mL) group, Glycyrrhizae with Aconitum (0.6 g / mL) group and Captopril (1.25 mg / mL) group. Each treatment group was given 5 mL / kg / D by gavage for 4 weeks, the normal group and model group were given the same volume of distilled water by gavage. (A) The content of ATP in left ventricular myocardium of rats determined by HPLC. (B) The content of AMP in left ventricular myocardium of rats determined by HPLC. (C) The content of ADP in left ventricular myocardium of rats determined by HPLC. Compared with the normal group, **P<0.01; ***P<0.001; Compared with the model group, #P<0.05; ##P<0.01; ###P<0.001.

Aconitum group was increased, but there was no significant difference with the model group, and the ATP level in the myocardial tissue of the compatibility of Glycyrrhizae and Aconitum group was significantly higher than that of the model group (0.03156 ± 0.006163; P < 0.001) (Fig 8A). Compared with the normal group, the content of AMP in left ventricular myocardium of CHF model group increased significantly (0.02345 ± 0.006343; P < 0.01); the level of AMP in the compatibility of Glycyrrhizae and Aconitum group decreased, but there was significant difference compared with the model group (-0.01456 ± 0.005776; P < 0.05) (Fig 8B). Compared with the normal group, the left ventricular ADP level of the model group was increased (0.09689 ± 0.009806; P < 0.05); after the intervention of Aconitum, the compatibility of Glycyrrhizae and Aconitum and captopril, the ADP level was lower than that of the model group, there was significant difference (-0.0557 ± 0.01849; -0.07513 ± 0.0137; -0.07185 ± 0.02716; P < 0.05; P < 0.001; P < 0.05) (Fig 8C). It can be seen that the effect of the compatibility of Glycyrrhizae and Aconitum on improving energy metabolism of CHF rats is better than that of Aconitum alone.

## Discussion

CHF is a common and frequently occurring disease. Over the years, TCM has been exploring in the treatment of CHF. This study analyzed the mechanism of Glycyrrhiza promoted the

metabolism of toxic components of Aconitum and relieved the development of CHF by compatibility with Aconitum.

Hepatocytes express a group of highly specific cytochrome P450 (CYP450) biotransferases, which can transform many drugs into inactive or hepatotoxic compounds [17]. CYP450 is the most important phase I drug metabolism enzyme system involved in the metabolism and transformation of various exogenous and endogenous substances in vivo [14, 18]. The main subtypes of CYP1A2, CYP2C, CYP2D6, CYP3A4/5 and CYP2E1 are involved in the metabolism of about 90% drugs in vivo [19]. This study showed that ketoconazole had the strongest inhibitory effect on the metabolism of AC, MA and HA in rat liver microsomes, and suggested that they were mainly metabolized by CYP3A. CYP3A is rich in liver and intestine, which is the most abundant subtype in CYP450 family, and participates in about 60% of drug metabolism [20]. When drugs are combined with CYP3A inhibitors or inducers in clinic, they may affect their own metabolism, slow down / accelerate their metabolic rate, increase / decrease their levels, and increase / decrease their toxicity in vivo.

Glycyrrhiza has the function of harmonizing various drugs and detoxifying. Aconitum is often used in combination with Glycyrrhiza to reduce toxicity. Does Glycyrrhiza promote the metabolism of toxic components of Aconitum by up regulating the expression of CYP3A. It has been reported that glycyrrhizic acid in Glycyrrhiza can activate pregnane X receptor (PXR) through a variety of ways, resulting in a concentration dependent up regulation of CYP3A 1 / 2 gene and protein expression [21]. In this study, the induction of CYP3A by Glycyrrhiza was studied at the whole animal level. PCR results showed that Glycyrrhiza could significantly increase the expression of CYP3A 1 / 2 mRNA. This study was consistent with the results of Paolini et al. [21]. In the later stage, we will further study the intervention effect of Glycyrrhiza on the toxicity of aconitum through SD rats. Since Glycyrrhiza had a clear induction effect on CYP3A expression, it would inevitably accelerate the metabolism of toxic components of Aconitum. In addition, the current research on the toxicity mechanism of the compatibility of Glycyrrhizae and Aconitum mainly focused on the changes of the dissolution of toxic components in the decoction before and after the compatibility of Glycyrrhizae and Aconitum. In this study, we detected the changes of metabolic stability parameters ($t_{1/2}$ and $CL_{int}$) of diester alkaloids in rat liver microsomes in each group (S1 and S2 Tables) and found that the metabolic rates of AC, MA and HA induced by different doses of Glycyrrhiza were different (medium dose group > low dose group > high dose group), the medium concentration group promoted the metabolism of Aconitum diester alkaloids in vivo better than the low concentration group, when compared with the high concentration, and there was no significant difference between high dose group and negative control group (S3 Table). It was possible that the induction effect of this concentration was in a state of satiation, and the specific reasons needed to be further explored. The results showed that Glycyrrhiza could accelerate the metabolism of Aconitum diester alkaloids in vivo, which mought be one of the mechanisms that Glycyrrhiza could reduce the toxicity and enhance the efficiency of Aconitum.

In traditional medicine, for patients with CHF, Aconitum is "the first drug of choice for invigorating Yang and saving lives". Modern pharmacological studies have confirmed that Aconitum has the effects similar to digitalis, such as positive muscle strength, dilating of coronary artery, anti myocardial ischemia and anti shock, but it can easily lead to cause ectopic arrhythmia [4, 5]. Glycyrrhiza has the effects of anti arrhythmia and protecting myocardial cells [8, 12]. These effects coincide with the principles of modern medicine in the treatment of heart failure (such as cardiotonic, vasodilator, etc.). Studies have confirmed that [22–24] BNP levels also increased with the gradual aggravation of heart failure. AngIIalso participates in the pathophysiological process of CHF formation, and is related to the severity of CHF [25–28]. At the same time, AngIIcan stimulate the secretion of ALD. The increase of ALD release can

cause arrhythmia, induce and aggravate heart failure. The results showed that the levels of BNP, Ang II and ALD in serum of CHF model group were significantly higher than those of normal group, while the above indexes of the compatibility of Glycyrrhizae and Aconitum group were significantly lower than those of model group and better than those of Aconitum alone. Myocardial energy metabolism disorder is an important pathological link in CHF. In case of CHF, the imbalance of oxygen supply and consumption, mitochondrial dysfunction, and the reduction of ATP production and utilization in myocardium aggravate the development of CHF [29]. It has been reported that CHF induced by ADR may be related to mitochondrial damage and energy imbalance [30]. The results showed that the content of ATP decreased and the content of ADP and AMP increased in CHF model group compared with the normal group, which indicated that the disorder of energy metabolism was involved in the pathological evolution. At the same time, compatibility of Glycyrrhizae and Aconitum could increase the content of ATP and reduce the content of AMP and ADP in CHF rats, which was better than Aconitum alone.

## Conclusion

This study focused on how Glycyrrhiza relieved the toxicity of Aconitum. Firstly, it was confirmed that Glycyrrhiza could induce the expression of CYP3A 1 / 2 mRNA. Secondly, it was verified that Glycyrrhiza could promote the metabolism of diester Aconitum alkaloids, and the compatibility of Glycyrrhiza and Aconitum reduced the exposure of toxic components in vivo. Finally, CHF model in rats showed that the protective effect of the compatibility of Glycyrrhiza and Aconitum on CHF was better than that of Aconitum alone. Glycyrrhiza could alleviate the toxicity of Aconitum and increased the protective effect of Aconitum on the heart.

## Supporting information

**S1 Table. In vitro $t_{1/2}$ data of AC, MA and HA in RLMs of pretreated rats.**
(DOCX)

**S2 Table. In vitro intrinsic clearance ($CL_{int}$) of AC, MA and HA in RLMs of pretreated rats.**
(DOCX)

**S3 Table. Metabolites A1, A2, M1 and H1 in vivo.**
(DOCX)

**S1 Fig.**
(TIF)

**S1 Data.**
(XLS)

## Acknowledgments

We want to thank Stefano Nembrini for his useful comments on our methodology section. We would also like to thank the editors of PLoS ONE and the reviewers for their thoughtful comments that helped us strengthen this article.

## Author Contributions

**Conceptualization:** Jian Jiang.

**Data curation:** Ping Miao, Jiangan Li, Su Tu.

**Formal analysis:** Jian Jiang.

**Funding acquisition:** Lulu Ni.

**Investigation:** Fang Wan.

**Methodology:** Lulu Ni, Ping Miao.

**Project administration:** Fang Wan.

**Software:** Min Ai.

**Supervision:** Min Ai.

**Validation:** Lingzhong Kong.

**Visualization:** Lingzhong Kong.

**Writing – original draft:** Lulu Ni.

**Writing – review & editing:** Su Tu.

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
