## [Decision Letter · Decision Letter 0]

14 Apr 2022

PONE-D-22-05670Glycyrrhiza uralensis promote the metabolism of toxic components of Aconitum carmichaeli by CYP3A and alleviate the development of chronic heart failurePLOS ONE

Dear Dr. Su Tu,

Thank you for submitting your manuscript to PLOS ONE. After careful consideration, we feel that it has merit but does not fully meet PLOS ONE’s publication criteria as it currently stands. Therefore, we invite you to submit a revised version of the manuscript that addresses the points raised during the review process.

We look forward to receiving your revised manuscript.

Kind regards,

Cecilia Zazueta, Ph. D.

Academic Editor

PLOS ONE

Journal Requirements:

2. PLOS requires an ORCID iD for the corresponding author in Editorial Manager on papers submitted after December 6th, 2016. Please ensure that you have an ORCID iD and that it is validated in Editorial Manager. To do this, go to ‘Update my Information’ (in the upper left-hand corner of the main menu), and click on the Fetch/Validate link next to the ORCID field. This will take you to the ORCID site and allow you to create a new iD or authenticate a pre-existing iD in Editorial Manager. Please see the following video for instructions on linking an ORCID iD to your Editorial Manager account: https://www.youtube.com/watch?v=_xcclfuvtxQ.

Additional Editor Comments:

It is recommended to performe an adequate statistical analysis and to arrenge the format of the manuscript, to make easier to review.

Reviewers' comments:

Reviewer's Responses to Questions

**Comments to the Author**

1. Is the manuscript technically sound, and do the data support the conclusions?

Reviewer #1: Yes

2. Has the statistical analysis been performed appropriately and rigorously? 

Reviewer #1: No

3. Have the authors made all data underlying the findings in their manuscript fully available?

Reviewer #1: Yes

4. Is the manuscript presented in an intelligible fashion and written in standard English?

Reviewer #1: No

5. Review Comments to the Author

Reviewer #1: In this manuscript, the metabolism of AC, MA and HA in Aconitum was intervened by regulating CYP450 with Glycyrrhiza, and the protective effect and mechanism of Glycyrrhiza combined with Aconitum on CHF were explored from the perspective of neuroendocrine system and myocardial energy metabolism in rats with heart failure, providing an in-depth basis for the pharmacological study of the compatibility of Aconitum carmichaeli with licorice. So I think that the manuscript is innovative and can be published in this journal.

Before this manuscript is accepted, there are some errors that need to be amended.

1、In the statistical analysis, the number of samples ( n ) is not well pointed out, which makes it difficult to ensure the statistical significance of the data.

2、In Figs 1B and 1D, there are still two unmarked peaks. why they are not explored?

3、In the discussion section, the sentence “we found that the metabolic rates of AC, MA and HA induced by different doses of Glycyrrhiza were different (medium dose group > low dose group > high dose group)” was pointed out. Therefore, intergroup comparisons should be added in statistical analysis.

4、The format of the manuscript is chaotic and needs to be corrected. Image clarity is not enough, it is difficult to see the data, such as Fig 3, Fig 6 and so on .

6. PLOS authors have the option to publish the peer review history of their article (what does this mean?). If published, this will include your full peer review and any attached files.

Reviewer #1: **Yes: **Xian-Ju Huang

---

## [Author Response · Author response to Decision Letter 0]

11 May 2022

May 6, 2022

Dear Reviewer: 

First of all, we thank you for your thoughtful suggestions and insights. The manuscript has benefited from these insightful suggestions. The manuscript ID is PONE-D-22-05670.

The manuscript has been rechecked and the necessary changes have been made in accordance with the reviewers’ suggestions. The responses to all comments have been prepared and included in the main text. 

Reviewer 1

EVALUATION: In this manuscript, the metabolism of AC, MA and HA in Aconitum was intervened by regulating CYP450 with Glycyrrhiza, and the protective effect and mechanism of Glycyrrhiza combined with Aconitum on CHF were explored from the perspective of neuroendocrine system and myocardial energy metabolism in rats with heart failure, providing an in-depth basis for the pharmacological study of the compatibility of Aconitum carmichaeli with licorice. So I think that the manuscript is innovative and can be published in this journal.

Before this manuscript is accepted, there are some errors that need to be amended.

1.In the statistical analysis, the number of samples ( n ) is not well pointed out, which makes it difficult to ensure the statistical significance of the data.

2.In Figs 1B and 1D, there are still two unmarked peaks. why they are not explored?

3.In the discussion section, the sentence “we found that the metabolic rates of AC, MA and HA induced by different doses of Glycyrrhiza were different (medium dose group > low dose group > high dose group)” was pointed out. Therefore, intergroup comparisons should be added in statistical analysis.

4.The format of the manuscript is chaotic and needs to be corrected. Image clarity is not enough, it is difficult to see the data, such as Fig 3, Fig 6 and so on .

Response: Thank you for your advice.

1.In the statistical analysis, n=3 in vitro, n=5 in vivo.

2.The two unmarked peaks in figures 1B and 1D contained both blank control samples and standard samples. They are considered to be impurities and did not affect the final result. 

3.In the study on the metabolic rate of AC, MA and HA induced by licorice, we further detected the changes of metabolic stability parameters (t1/2 and CLint) of diester alkaloids in rat liver microsomes in each group (Supplementary Table 1 and 2) and carried out statistical analysis between groups. At the same time, the differences in the production of A1, A2, M1 and H1 metabolites in different groups were statistically analyzed (Supplementary Table 3). 

4.Thank you for your advice. We have corrected the format of the manuscript. Because the columns in Figure 3 and the lines in Figure 6 were dense, we chose to upload them in the form of a single graph, so that you could clearly see all the datas in the graph. In Figure 5, the trend line color of each group was changed to make it clearer. And other pictures could be enlarged to see clearly. 

All the above changes will be presented at the revised.

Thank you for your consideration. I look forward to hearing from you.

Sincerely,

Dr. Su Tu

Department of emergency, the Affiliated Wuxi NO.2 People's Hospital of Nanjing Medical University, No. 68, Zhongshan Road, Wuxi, Jiangsu 214000, PR China

Tel: +86-0510-68562222

Email: tusuwxey@126.com

---

## [Decision Letter · Decision Letter 1]

3 Jun 2022

Glycyrrhiza uralensis promote the metabolism of toxic components of Aconitum carmichaeli by CYP3A and alleviate the development of chronic heart failure

PONE-D-22-05670R1

Dear Dr. Su Tu,

We’re pleased to inform you that your manuscript has been judged scientifically suitable for publication and will be formally accepted for publication once it meets all outstanding technical requirements.

Kind regards,

Cecilia Zazueta, Ph. D.

Academic Editor

PLOS ONE

Additional Editor Comments (optional):

Reviewers' comments:

Reviewer's Responses to Questions

**Comments to the Author**

1. If the authors have adequately addressed your comments raised in a previous round of review and you feel that this manuscript is now acceptable for publication, you may indicate that here to bypass the “Comments to the Author” section, enter your conflict of interest statement in the “Confidential to Editor” section, and submit your "Accept" recommendation.

Reviewer #1: All comments have been addressed

2. Is the manuscript technically sound, and do the data support the conclusions?

Reviewer #1: Yes

3. Has the statistical analysis been performed appropriately and rigorously? 

Reviewer #1: Yes

4. Have the authors made all data underlying the findings in their manuscript fully available?

Reviewer #1: Yes

5. Is the manuscript presented in an intelligible fashion and written in standard English?

Reviewer #1: Yes

6. Review Comments to the Author

Reviewer #1: (No Response)

7. PLOS authors have the option to publish the peer review history of their article (what does this mean?). If published, this will include your full peer review and any attached files.

Reviewer #1: **Yes: **Xian-Ju Huang

---

## [Editor Report · Acceptance letter]

17 Jun 2022

PONE-D-22-05670R1 

Glycyrrhiza uralensis promote the metabolism of toxic components of Aconitum carmichaeli by CYP3A and alleviate the development of chronic heart failure 

Dear Dr. Tu:

I'm pleased to inform you that your manuscript has been deemed suitable for publication in PLOS ONE. Congratulations! Your manuscript is now with our production department. 

Kind regards, 

on behalf of

Dr. Cecilia Zazueta 

Academic Editor

PLOS ONE